# Factors Contributing to Chronic Kidney Disease following COVID-19 Diagnosis in Pre-Vaccinated Hospitalized Patients

**DOI:** 10.3390/vaccines11020433

**Published:** 2023-02-13

**Authors:** Diana Laila Ramatillah, Michael Michael, Kashifullah Khan, Nia Natasya, Elizabeth Sinaga, Silvy Hartuti, Nuzul Fajriani, Muhammad Junaid Farrukh, Siew Hua Gan

**Affiliations:** 1Faculty of Pharmacy, Universitas 17 Agustus 1945 Jakarta, North Jakarta 14350, Indonesia; 2Department of Clinical Pharmacy, College of Pharmacy, University of Hail, Hail 2440, Saudi Arabia; 3Faculty of Pharmaceutical Sciences, UCSI University, Kuala Lumpur 56000, Malaysia; 4School of Pharmacy, Monash University Malaysia, Jalan Lagoon Selatan, Bandar Sunway 47500, Malaysia

**Keywords:** CKD, COVID-19, survival rate, ICU

## Abstract

In this study, we aim to evaluate the factors that may contribute to the development of chronic kidney disease following COVID-19 infection among hospitalized patients in two private hospitals in Jakarta, Indonesia. This is a retrospective cohort study between March 2020 and September 2021. Patient selection was conducted with a convenience sampling. All patients (n = 378) meeting the inclusion criteria during the study period were enrolled. Various sociodemographic, laboratory test, and diagnostic parameters were measured before the determination of their correlation with the outcome of COVID-19 infection. In this study, all pre-vaccinated patients with COVID-19 had no history of chronic kidney disease (CKD) prior to hospital admission. From this number, approximately 75.7% of the patients developed CKD following COVID-19 diagnosis. Overall, significant correlations were established between the clinical outcome and the CKD status (*p* = 0.001). Interestingly, there was a significant correlation between serum creatinine level, glomerular filtration rate (GFR), and CKD (*p* < 0.0001). Oxygen saturation (*p* = 0.03), admission to the intensive care unit (ICU) (*p* < 0.0001), and sepsis (*p* = 0.005) were factors that were significantly correlated with CKD status. Additionally, the type of antibiotic agent used was significantly correlated with CKD (*p* = 0.011). While 82.1% of patients with CKD survived, the survival rate worsened if the patients had complications from hyperuricemia (*p* = 0.010). The patients who received levofloxacin and ceftriaxone had the highest (100%) survival rate after approximately 50 days of treatment. The patients who received the antiviral agent combination isoprinosine + oseltamivir + ivermectin fared better (100%) as compared to those who received isoprinosine + favipiravir (8%). Factors, such as hyperuricemia and the antibiotic agent used, contributed to CKD following COVID-19 hospitalization. Interestingly, the patients who received levofloxacin + ceftriaxone and the patients without sepsis fared the best. Overall, patients who develop CKD following COVID-19 hospitalization have a low survival rate.

## 1. Introduction

In December 2019, the frequency of pneumonia with unknown cause started to rise in the province of Hubei, Wuhan China [1,2]. The world was shocked by this new virus causing pneumonia called Severe Acute Respiratory Syndrome 2 (SARS-CoV-2), which became a worldwide outbreak. On 19 January 2022, approximately 335 million cases were confirmed with 5.5 million fatalities. SARS-CoV-2 spreads via multiple hosts, culminating in several symptoms, including common cold-like symptoms to a severe and sometimes deadly respiratory infection [2,3,4]. The virus enters the body through the respiratory route and the host’s cells through a cell receptor.

An important way to combat the spread of COVID-19 infection is vaccination [5]. However, in Indonesia, vaccination only started in the middle of January 2021, first applied to only health workers [6]. COVID-19 vaccination for the public started in early July 2021, although patients with comorbidities, such as chronic kidney disease (CKD), autoimmune disease, and cancer as well as pregnant women, were not even recommended for early vaccination [7,8].

Advanced age, immunosuppressive therapy, and underlying comorbidities, including cardiovascular and chronic pulmonary illnesses, diabetes, cancers, and CKD, are contributing factors that cause a higher risk of the severe clinical presentation in patients with COVID-19 [9,10,11,12]. Impaired renal function is commonly observed during systemic sepsis [13]. Depending on the severity of the infection and the organism responsible [14], renal involvement may vary from having insignificant proteinuria to having acute kidney injury that requires dialysis. It is interesting to note that some individuals suffering from severe cases of COVID-19 may also show signs of kidney damage, although they may not have any underlying kidney problems prior to the infection [15,16,17].

Direct microbial invasion of the renal tissues or in the collecting system may cause some renal involvement [14]. Activation of the immunologic pathways or immune complexes resulting from the infection process may also damage the kidneys [14]. Therefore, the possibility of drug-induced nephrotoxicity caused by antimicrobial therapy should always be considered in the evaluation of renal complications occurring among patients with infectious diseases [14]. Acute kidney injury (AKI), which is more common among patients with severe disease and particularly in patients in the intensive care unit (ICU), is deemed a negative prognostic factor for survival [17,18,19]. In a retrospective study on patients with both COVID-19 and pneumonia (n = 333), those with kidney dysfunction had higher mortality rates than those without kidney involvement (11.2 and 1.2%, respectively) [19,20].

CKD is a condition where there is reduced kidney function due to the inability of the organ to filter blood efficiently [21]. Initial findings reported that patients with chronic conditions including CKD had a greater preponderance of developing severe COVID-19 [22,23]. Based on the OpenSAFELY database, there is a graded relationship between kidney disease and COVID-19 mortality rate. The report indicates that individuals with severe forms of CKD have a significantly higher risk of COVID-19 mortality than other recognized high-risk categories, including hypertension, obesity, chronic heart disease, and lung disease [24]. In fact, patients with CKD have a 14- to 16-fold higher mortality rate due to lung infection [25] when compared to the general population.

Currently, the treatment for CKD involves the blockade of the renin–angiotensin–aldosterone system (RAAS). However, there is some debate about whether or not to deploy the RAAS blockade in the context of COVID-19. SARS-CoV-2, similar to the SARS-CoV-1 virus that first appeared in 2003, enters the body through the angiotensin-converting enzyme 2 (ACE2) receptor. This phenomenon is alarming, considering that ACE2 is found in several organs, including lung epithelial type-2 alveolar cells and renal tubular epithelial cells [21,22]. ACE1 converts angiotensin I to angiotensin II, whereas ACE2 converts angiotensin II to angiotensin, both of which operate on the mitochondrial assembly protein (MAS) receptor found in numerous body tissues.

Since the counter-regulatory system to vasoconstriction is contributed by angiotensin-II binding to the angiotensin 1 receptor, vasodilation ensues, causing reduced systemic inflammation [26]. The potential negative impact of continued ACEI or angiotensin 2 receptor blocker (ARB) use, including the potential up-regulation of the ACE2 receptor that may boost the virus’ ability to penetrate cells, remains a debate. It is plausible that ACEIs aid by inhibiting viral entry via blockage of the ACE2 receptor. In recent months [23,24,26], some bigger studies investigated the possible link between the use of ACEIs or ARBs and negative outcomes, including being COVID-19 positive and high mortality rate, with no conclusive evidence observed.

SARS-CoV-2 infects the kidneys and may trigger an AKI, although the effects on the kidney in patients with underlying kidney disease are not well-characterized [19]. To date, it is still unknown if the stages of CKD are correlated with COVID-19 outcomes, necessitating more research on this topic. To our knowledge, this is the first attempt to correlate CKD status with the outcome of COVID-19 in Indonesia.

## 2. Materials and Methods

### 2.1. Study Design and Setting

This is a cohort retrospective study based on the patient’s complete medical record conducted at two private hospitals in Jakarta. The patients were screened against the inclusion/exclusion criteria. The inclusion criterion was all adult patients with COVID-19. The exclusion criteria were patients with a history of CKD prior to hospital admission (stages 2–5), pregnant patients, patients with autoimmune diseases, patients with cancer, and patients with human immunodeficiency viruses (HIV)/AIDS. Patients with positive RT-PCR COVID-19 tests were recruited via convenience sampling between March 2020 and September 2021. Patients who were diagnosed as having CKD following confirmation of COVID-19 status were included in the study. The GFR was calculated based on the CKD Epidemiological Collaboration (CKD-EPI), while the CKD grade was determined with KDIGO as indicated by the National Kidney Foundation (Figure 1). CKD was further confirmed via the determination of blood urea nitrogen (BUN) and serum creatinine levels. The sociodemographic, laboratory test (following COVID-19 diagnosis), and diagnostic data were collected before being transcribed into the clinical research forms (CRF).

The descriptive data were analyzed using the Mann–Whitney, Wilcoxon, chi-squared, Cox regression, and Kaplan–Meier tests using SPSS software version 26.0. A significant correlation was indicated by a *p*-value < 0.05.

### 2.2. Ethical Approval

Ethical approvals were obtained from two ethical medical committees, (No.01/KEPK-UTA45JKT/EC/EXE/11/2021) and (No.14/KEPK-UTA45JKT/EC/EXE/12/2021), at the study sites prior to data collection. The ethical approvals complied with the Declaration of Helsinki.

### 2.3. Data Collecting and Handling

The data were arranged according to the sociodemographic status, laboratory tests following COVID-19 diagnosis, and current medication and transcribed to the CRF.

### 2.4. Definitions

Anemia = hemoglobin (Hb) concentration and/or red blood cell (RBC) numbers, which are lower than normal and are insufficient to meet the physiological needs of an individual [28,29].

Clinical outcome = the final result of the treatment (death or recover).

Heart disease complication = as per the doctor’s diagnosis based on the medication history, history of diseases, and the echocardiograms (ECG) [30].

Hypertension = a blood pressure (BP) more than 140/90 mmHg [30,31].

Hypercholesterolemia = total cholesterol more than 200 mg/dL, or having a low-density lipoprotein (LDL) of more than 100 mg/dL, or triglyceride levels of more than 150 mg/dL, or HDL less than 40 mg/dL [32].

Hyperuricemia = the presence of high levels of uric acid in the blood [33].

Sepsis = a medical emergency that describes the body’s systemic immunological response to an infectious process that may lead to the development of end-stage organ dysfunction and death [34].

## 3. Results

### 3.1. Associations between CKD Status and Sociodemographic Profiles among Patients with COVID-19

In this study, all of the pre-vaccinated patients with COVID-19 (n = 378) had no history of being diagnosed with CKD prior to hospital admission. CKD is defined as having renal failure between stages 2 and 5 (or having a creatinine clearance (CrCL) of less than 90 mL/min). In the study, five patients required dialysis. The majority (75.7%) were later diagnosed as having CKD (Figure 2). There was a significant correlation between oxygen saturation, serum creatinine level, and glomerular filtration rate (GFR) with CKD (*p* < 0.05). In terms of gender, 77.16% of females developed CKD as compared to males (74.03%). The patients who developed CKD had a median serum creatinine level and GFR of 1.2 mg/dL and 57.8 mL/min, respectively (Table 1).

### 3.2. Association between Patients’ CKD Status and Diagnosis

There was a significant correlation between (1) hospitalization status and (2) the overall clinical outcome in patients with COVID-19 with their CKD status (*p* < 0.05) (Table 2). Unfortunately, 155 patients passed away with more than half (66.5%) of the fatalities being patients with CKD.

According to the WHO, there are three categories for the stage of COVID-19: [1] critical COVID-19—in this category, patients have already developed acute respiratory distress syndrome (ARDS), sepsis, septic shock, or the condition that normally requires the provision of life-sustaining therapies; [2] severe COVID-19—oxygen saturation <90% on room air with signs of pneumonia and signs of severe respiratory distress; and [3] non-severe COVID-19—the absence of the signs of severe or critical disease [35]. However, according to the health minister of the republic of Indonesia, there are four categories of COVID-19 severity: [1] asymptomatic, [2] moderate symptoms (patient with pneumonia symptoms and SpO2 93–95%), [3] severe symptoms (patient with pneumonia symptoms and SpO2 <93%), and [4] critical illness (patient with Acute Respiratory Distress Syndrome (ARDS), sepsis, and septic shock [36]. In this study, the disease severity is based on the local guideline from the Ministry of Health of Indonesia.

### 3.3. Association between CKD Status and Laboratory Tests

Almost all patients had D-dimer values of more than 500 mg/L FEU (Table 3).

### 3.4. Contributing Factors to CKD following COVID-19 Diagnosis

Factors, such as hyperuricemia and antibiotic use, were significantly correlated with CKD status post-COVID-19 infection (Table 4). In fact, having hypertension increased the propensity for CKD by more than 60%.

In this study, sepsis and hospitalization status contributed to the higher mortality rate among patients with COVID-19 with both showing a higher significance on the probability of dying (approximately 30%) (Table 5).

Favipiravir was the most used antiviral (19.6%), while levofloxacin was the most common antibiotic in this study (Appendix A).

A 7% survival rate after approximately 90 days of treatment was observed among the patients who developed CKD in the hospital (Figure 3). In this study, the survival rate among patients with COVID-19 who received isoprinosine + favipiravir after approximately 79 days of treatment was 8%, but for those patients who received isoprinosine + oseltamivir + ivermectin, the survival rate after approximately 18 days of treatment was 100%. Patients who received isoprinosine + remdesivir + favipiravir had an approximately 48% survival rate after approximately 50 days of treatment (Figure 4). However, patients who received levofloxacin + ceftriaxone had the highest (100%) survival rate after approximately 50 days of treatment, but only a survival rate of 50% after approximately 79 days of treatment (Figure 5).

## 4. Discussion

In this study, most patients who developed CKD had a serum creatinine level of 1.2 mg/dL and GFR of 57.8 mL/min; a significant correlation between serum creatinine level and glomerular filtration rate (GFR) with CKD was established. Based on the KDIGO, the normal GFR is 90 to 120 mL/min [27].

Most of the hospitalized patients were of an advanced age (approximately 55 years old). Therefore, they tended to have weaker immune systems as well as some comorbidities, such as diabetes mellitus, hypertension, and heart disease complications [4,35]. These phenomena also contributed to them having a more severe COVID-19 infection, leading to ICU admission [36] as evidenced in our study where 32.8% were treated in the ICU.

Based on the patients’ medical records, 286 patients were diagnosed with CKD following COVID-19 diagnosis. Additionally, it is reported that there is a graded association between the level of kidney dysfunction and the risk of mortality from COVID-19 [9]. Other studies indicate that COVID-19 infection can further decrease the GFR in patients with CKD (mean = 3.57 min/mL) [37]. Lin et al. reported that acute kidney injury was a common and serious complication of COVID-19 [38]. Since coronavirus is reported to enter the cells by binding to the ACE2 receptor, the high level of ACE2 receptors found in the kidneys [39,40,41] indicates that the kidney cells may be the point of entry for SARS-CoV-2 to affect the renal function.

Our data indicate that most patients with COVID-19 who developed CKD were also diagnosed with diabetes mellitus (75.9%). In fact, a poorer outcome for patients with COVID-19 was associated with diabetes mellitus and was also influenced by factors such as age and hypertension [42]. Another risk factor that facilitates the development of severe to a more critical form of COVID-19 is diabetes [43]. Patients with CKD who are at a high risk of having hyperglycemia and hypoglycemia have dramatically disturbed glucose homeostasis. In a study on a group of patients, high and low glycemic rates were linked to higher morbidity and a shorter survival rate [44]. Overall, reduced renal gluconeogenesis compromised metabolic pathways (including altered medication metabolism) and decreased insulin clearance are linked to a higher risk of hypoglycemia in patients with CKD. On the other hand, inflammation-induced insulin resistance and reduced glucose filtration as well as excretion are predisposing factors for hyperglycemic episodes [44].

As expected, urea level is significantly higher in patients with CKD post-COVID-19. In CKD, blood urea nitrogen (BUN) levels are markedly increased, reaching a height in patients with end-stage renal disease [45]. Additionally, a significant correlation is established between chloride ions and patients’ pre- and post-CKD statuses. In two of the studies, a worsened renal outcome was reported with the critical and non-critically ill patients who received intravenous fluids having higher chloride concentrations [46,47]. This phenomenon is observed because nephron damage in patients with CKD tends to cause an inability of the kidneys to maintain fluid and electrolyte homeostasis [48].

Based on the medical records, the patients developed sepsis from bacteremia, which progressed to SIRS (Systemic Inflammatory Response Syndrome). In our study, most patients had sepsis [49] as indicated by the presence of two Systemic Inflammatory Response Syndrome (SIRS) symptoms (fever and high WBC) and a confirmed infection as reported by the doctor’s diagnosis and written in the medical record. There was a significant correlation (*p* < 0.05) between CKD statuses and clinical outcomes. There was also a positive correlation between sepsis (*p* < 0.0001) and the probability of dying in the affected patients. However, the mortality rate in the patients with both CKD and sepsis complications was very high (93%). Additionally, a systemic inflammatory response from COVID-19 infection can result in kidney injury and impairment of sodium and water metabolism, which may precipitate a worsening heart failure [45,46].

In this study, hyperuricemia and antibiotic use contributed to CKD following COVID-19 diagnosis. Based on the study by Chauhan et al., uric acid (UA) was associated with increased AKI (odds ratio [OR] 2.8, 95% confidence interval [CI] 1.9–4.1) and in-hospital mortality (OR 1.7, 95% CI 1.3–2.3) [50].

A previous study conducted in Wuhan, China reported that 38.5% of patients (n = 226) died in the critical care unit [51], while in Africa, the mortality rate was as high as 54.7% within 30 days of ICU admission [52]. The patients who tended to have low survival rates were those admitted to the ICU 24 h before hospitalization for COVID-19, indicating that the patients who came to the hospital with a critical illness tended to have a lower survival rate as compared to the patients who came to the hospital with mild to moderate infections. Therefore, support could directly be given to the patients who were already admitted to the hospital, although patients in critical condition may also need to wait for a room to be available in a third-world country such as Indonesia.

Administration of isoprinosine and favipiravir yielded a low (8%) survival rate outcome with a mean duration of hospitalization of 79 days. To our knowledge, no study on dose adjustment for favipiravir as an antiviral treatment for patients with both CKD and COVID-19 has been reported [53]. However, there is a similar report for oseltamivir, where its dose in patients with both CKD and a creatine clearance of less than 60 mL/min was reduced from 75 mg to only 30 mg (twice daily), while for patients with CrCL of less than 30 mL/min, the maximum dose of oseltamivir was further capped at 30 mg/day [52]. The drug tends to stay longer in the body of patients with CKD due to the inability of the kidneys to secrete the final product of metabolism or the parent drug itself.

Clinical trials exclude patients with severe renal impairment, but favipiravir shows efficacy in treating COVID-19 pneumonia, especially in some patients with end-stage renal disease (ESRD) who are on hemodialysis [54]. Another antiviral agent effective against COVID-19 is isoprinosine. In fact, in this study, a combination of isoprinosine, oseltamivir, and ivermectin showed the highest survival rate (100%) after approximately 18 days of treatment. Isoprinosine is an immunomodulatory, antiviral drug licensed since 1971 in several countries worldwide to treat viral infections [55]. It triggers a non-specific immune response independent of the specific viral antigen responsible for influenza-like illness [56]. Therefore, overall, it is obvious that personalized treatment is recommended, especially in patients with both CKD and COVID-19 infection for maximized therapy. Favipiravir is an antiviral agent that selectively and potently inhibits the RNA-dependent RNA polymerase (RdRp) of RNA viruses [57].

Besides favipiravir, remdesivir is also used in treating COVID-19 infections. Although remdesivir is deemed to be a relatively nephrotoxic agent and usually is administered to patients with more severe COVID-19 infection, affected patients are more vulnerable to the development of AKI. Administration of remdesivir is not recommended in patients with an eGFR below 30 mL/min [58,59,60]. In some studies [59,60], using remdesivir for COVID-19 treatment was reported to increase the risk of developing AKI as compared to using other drugs, such as tocilizumab, hydroxychloroquine, and lopinavir/ritonavir. In fact, administration of remdesivir for more than five days was reported to worsen AKI [58,61], except for a single study by Izcovich et al. where remdesivir was reported to pose no significant risk for AKI when used as a COVID-19 treatment [62]. Overall, the active form of remdesivir acts as a nucleoside analog and inhibits the RNA-dependent RNA polymerase (RdRp) of coronaviruses including SARS-CoV-2 [63].

The type of antibiotics used is also important, especially in patients with both COVID-19 and CKD. To date, both levofloxacin (a third-generation fluoroquinolone) and ceftriaxone (a third-generation cephalosporin) have been promising in patients with CKD, although levofloxacin is reported to cause a rare form of neurotoxicity [56,64,65]. Additionally, macrolides such as azithromycin are one of the antibiotics used for COVID-19 [5]. Nevertheless, to date, the combination of levofloxacin (a third-generation fluoroquinolone) and ceftriaxone (a third-generation cephalosporin) have shown good synergism in treating pneumonia [66], although there are limited data on the type of antibiotic suitable for COVID-19. In our study, patients who received azithromycin had the highest (82%) survival rate after approximately 10 days of treatment, indicating that it is ideal.

## 5. Conclusions

Patients who have CKD following COVID-19 hospitalization status have a low survival rate. The type of antibiotic agent used was one of the contributing factors for CKD following COVID-19 among these patients. The combination of isoprinosine, oseltamivir, and ivermectin was highly effective in inhibiting the replication of the COVID-19 virus, while levofloxacin and ceftriaxone were the antibiotics of choice. Additionally, hyperuricemia contributed to CKD following COVID-19 hospitalization. Besides that, sepsis and hospitalization status (ICU) impacted mortality among these patients with both having equal mortality rates. Overall, patients with CKD should be carefully monitored to confer better protection against COVID-19 due to the higher mortality rate observed.

### 5.1. Limitation of Study

Our study has some limitations. The study duration was relatively short, and the samples were from two sites (both from private hospitals in Jakarta). The patient data retrieval is rather complicated in Indonesia since it is directly related to the quality of the data recording.

### 5.2. Future Study

Since the study was conducted among pre-vaccinated patients, future studies are needed to evaluate the kidney impact following vaccination.

## Figures and Tables

**Figure 1 vaccines-11-00433-f001:**
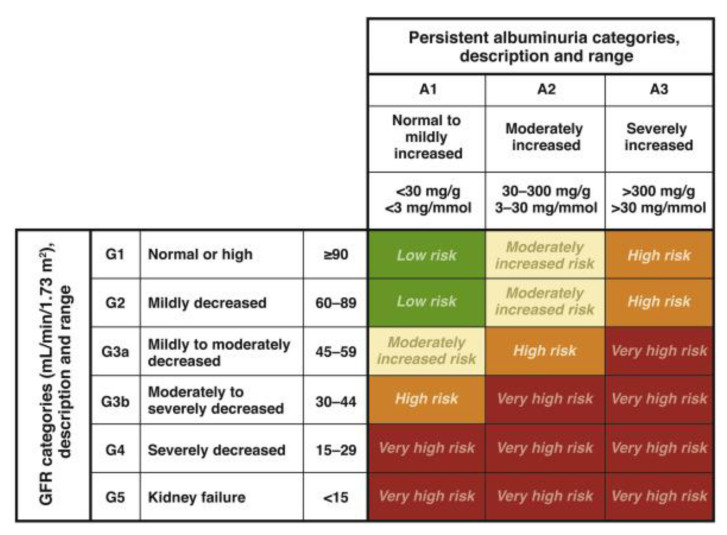
CKD grade was determined with KDIGO as indicated by the National Kidney Foundation [27].

**Figure 2 vaccines-11-00433-f002:**
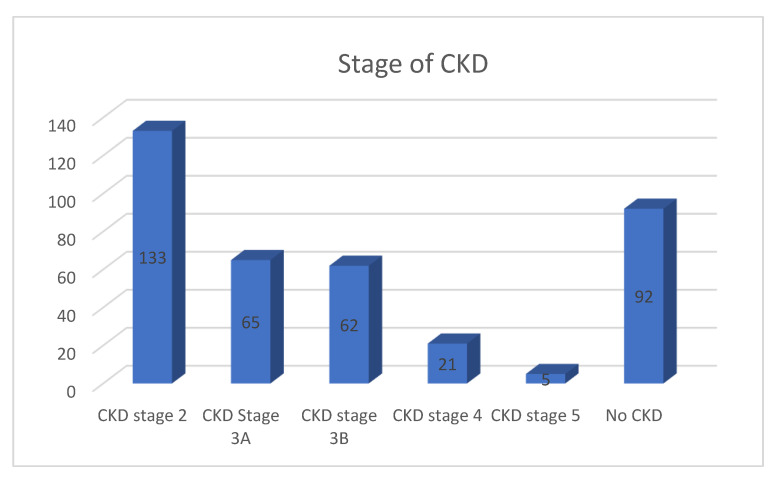
CKD grade following COVID-19 infection.

**Figure 3 vaccines-11-00433-f003:**
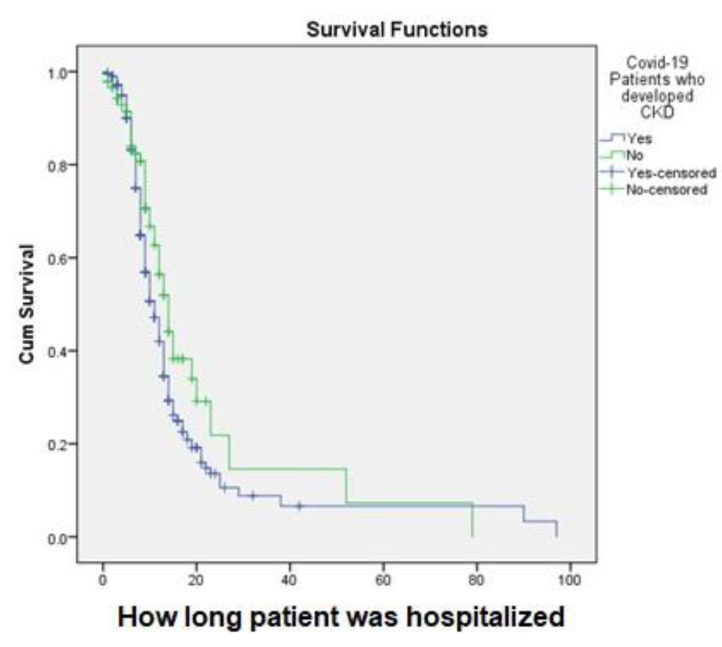
Kaplan–Meier survival analysis of the type of CKD.

**Figure 4 vaccines-11-00433-f004:**
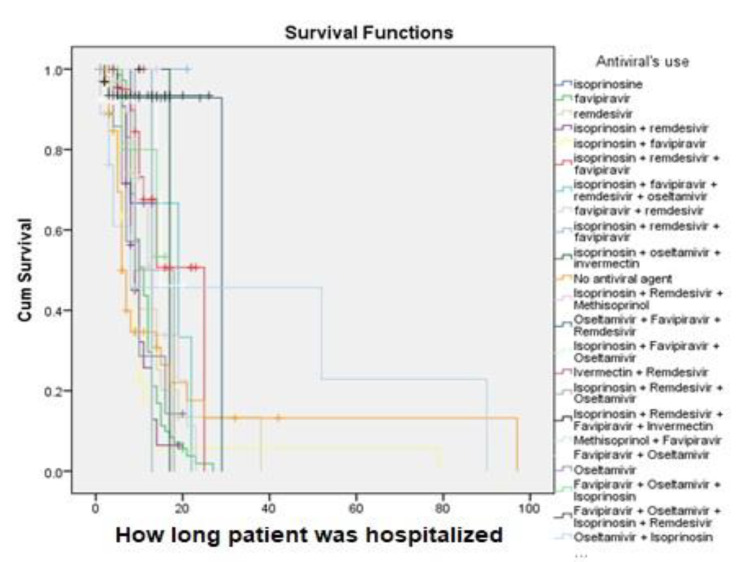
Kaplan–Meier survival analysis of the antiviral agents used.

**Figure 5 vaccines-11-00433-f005:**
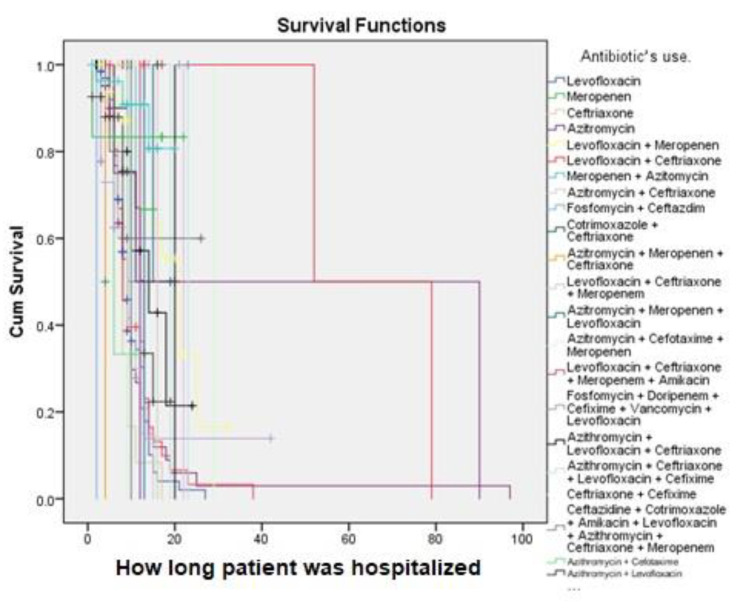
Kaplan–Meier survival analysis of the antibiotic used.

**Table 1 vaccines-11-00433-t001:** Association between CKD and the sociodemographic, vital signs, and kidney function parameters during hospital admission.

Factor	Total Patients with COVID-19 (n = 378) N (%)/Median	Patients with COVID-19 without CKD (n = 92) N (%)/Median	Patients with COVID-19 Who Developed CKD (n = 286) N (%)/Median	*p*-Value
Gender	Male = 181 (47.88%) Female = 197 (52.12%)	Male = 47 (25.97) Female = 45 (22.84)	Male = 134 (74.03) Female = 152 (77.16)	0.557 ^a^
Age	55	55	55	0.449 ^b^
Body mass index (BMI)	24.2	24.2	24.2	0.280 ^b^
Systolic/diastolic blood pressures (mmHg)	140.0	142.0	140.0	0.131 ^b^
Respiratory rate/min	24.0	24.5	24.0	0.144 ^b^
Temperature (°C)	37.0	36.8	57.0	0.638 ^b^
Pulse	98.0	98.0	98.0	0.972 ^b^
Oxygen saturation	93.0	90.5	94.0	0.03 ^b^
Blood urea nitrogen (BUN) mg/dL	25.0	27.5	25.0	0.513 ^b^
Serum creatinine (mg/dL)	1.2	0.8	1.2	<0.0001 ^b^
Glomerular filtration rate (GFR) (mL/min)	67.8	113.5	57.8	<0.0001 ^b^

^a^ = chi-squared test (N%), ^b^ = Mann–Whitney (Median). Post-CKD: patients diagnosed by the doctor after or during SARS-CoV-2 infection in the ward, and all patients who had a history of CKD prior to hospital admission were excluded.

**Table 2 vaccines-11-00433-t002:** Association between CKD status with diagnosis and clinical outcome.

Factor	Total Patients with COVID-19 (n = 378) N (%)	Patients with COVID-19 without CKD (n = 92) N (%)	Patients with COVID-19 Who Developed CKD (n = 286) N (%)	*p*-Value
Hospitalization status	ICU = 124 (32.8%) Non-ICU = 254 (67.2%)	ICU = 48 (38.7%) Non-ICU = 44 (17.3%)	ICU = 76 (61.3%) Non-ICU = 210 (82.7%)	<0.0001 ^a^
Sepsis	Yes = 121 (32.0%) No = 257 (67.9%)	Yes = 41 (33.9%) No = 52 (19.8%)	Yes = 80 (66.1%) No = 206 (80.2%)	0.005 ^a^
Diabetes mellitus	Yes = 129 (34.1%) No = 249 (65.9%)	Yes = 31 (24.0%) No = 61 (24.5%)	Yes = 98 (75.9%) No = 188 (75.5%)	1.000 ^a^
Hypercholesterolemia	Yes = 91 (24.1%) No = 287 (75.9%)	Yes = 25 (27.5%) No = 67 (23.3%)	Yes = 66 (72.5%) No = 220 (76.7%)	0.510 ^a^
Hypertension	Yes = 189 (50.0%) No = 189 (50.0%)	Yes = 41 (21.7%) No = 51 (26.9%)	Yes = 148 (78.3%) No = 138 (73.0%)	0.281 ^a^
Hyperuricemia	Yes = 132 (34.9%) No = 246 (65.1%)	Yes = 43 (32.6%) No = 49 (19.9%)	Yes = 89 (67.4%) No = 197 (80.1%)	0.009 ^a^
Anemia	Yes = 26 (6.9%) No = 352 (93.1%)	Yes = 4 (15.4%) No = 88 (25.0%)	Yes = 22 (84.6%) No = 264 (75.0%)	0.387 ^a^
Heart disease complication	Yes = 113 (29.9%) No = 265 (70.1%)	Yes = 24 (21.2%) No = 68 (26.6%)	Yes = 89 (78.8%) No = 197 (74.3%)	0.432 ^a^
Clinical Outcomes	Recovered = 223 (58.9%) Death = 155 (41.0%)	Recovered = 40 (17.9%) Death = 52 (33.5%)	Recovered = 183 (82.1%) Death = 103 (66.5%)	0.001 ^a^

^a^ = chi-squared test.

**Table 3 vaccines-11-00433-t003:** Association between CKD status with laboratory tests.

Parameter	Total Patients with COVID-19 (n = 378) Median	Patients with COVID-19 without CKD (n = 92) Median	Patients with COVID-19 Who Developed CKD (n = 286) Median	*p*-Value
Sodium (mmol/L)	135	134	135	0.309 ^a^
Potassium (mmol/L)	3.9	3.9	3.9	0.160 ^a^
Chloride (mmol/L)	101	99	101	0.194 ^a^
D-dimer (mg/L FEU)	755	890	710	0.089 ^a^

^a^ = Mann–Whitney (Median).

**Table 4 vaccines-11-00433-t004:** Contributing factors to CKD following COVID-19 diagnosis.

Factors	B	SE	Sig.*	Odds Ratio	95.0% CI for Odds Ratio	
					Lower	Upper
Gender	−0.081	0.123	0.513	0.923	0.725	1.175
Sepsis	0.195	0.152	0.198	1.215	0.903	1.635
Anemia	0.220	0.241	0.362	1.246	0.777	1.997
Hypertension	0.050	0.122	0.685	1.051	0.827	1.336
Hyperuricemia	−0.381	0.148	0.010	0.683	0.511	0.912
Diabetes	0.057	0.128	0.658	1.058	0.823	1.361
Antiviral agent	−0.012	0.011	0.253	0.988	0.968	1.009
Antibiotic use	−0.014	0.005	0.011	0.986	0.976	0.997
Hypercholesterolemia	0.029	0.215	0.894	1.029	0.675	1.569
Heart Disease Complication	0.273	0.235	0.246	1.313	0.828	2.083
Hospitalization status	−0.081	0.123	0.513	0.923	0.725	1.175

* Survival Cox regression analysis.

**Table 5 vaccines-11-00433-t005:** Contributing factors to the mortality rate in hospitalized patients with COVID-19.

Factor	B	SE	Sig.*	Odds Ratio	95.0% CI for Odds Ratio	
					Lower	Upper
Sepsis	−1.054	0.181	*p* < 0.0001	0.349	0.245	0.497
Heart Disease Complication	0.142	0.171	0.408	1.153	0.824	1.613
Hospitalization status	−1.203	0.181	*p* < 0.0001	0.300	0.211	0.428
CKD Status	−0.319	0.441	0.470	0.727	0.306	1.727
CKD Stage	0.071	0.097	0.467	1.073	0.887	1.299

* Survival Cox regression analysis.

## Data Availability

https://figshare.com/articles/dataset/Post_Covid-19_Kidney_Failure_In_Hospitalized_Pre-vaccinated_Patients_at_a_Private_Hospital_Jakarta/21335583.

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
