# Peer review of "Factors Contributing to Chronic Kidney Disease following COVID-19 Diagnosis in Pre-Vaccinated Hospitalized Patients"

_vaccines, 2023, doi:10.3390/vaccines11020433_

Round 1

Reviewer 1 Report (Previous Reviewer 2)

This amended version has been improved, as some clinical markers of nephropathy have been further developed. As such it is deemed for publication.

Author Response

Reviewer 2 Report (Previous Reviewer 3)

All requested comments and revisions have been implemented in the manuscript by authors. 

Some minor comments:

- The left column (p-value) in tables 4 & 5 seems unnecessary and it should be removed.

- Table 6 appear to be large and it can be transferred to the Supplementary.

- The spelling of medications throughout the manuscript should be checked and detected errors (e.g., Invermectin, isoprinosin) should be corrected.

- The first sentence of the discussion section should be rephrased.  

Author Response

Reviewer 3 Report (Previous Reviewer 1)

In this resubmitted/revised manuscript, the authors have earnestly attempted to address the critiques from the previous review. I think the revised manuscript is significantly improved in quality from the original. 

Author Response

This manuscript is a resubmission of an earlier submission. The following is a list of the peer review reports and author responses from that submission.

Round 1

Reviewer 1 Report

This is a retrospective study that sought to identify factors that contribute to chronic kidney disease in Covid-19 patients admitted to two hospitals in Indonesia. While the study sample size and design seem to be satisfactory, the manuscript can be improved by addressing the points mentioned below:

1) The authors should clarify the inclusion criteria. This statement is rather confusing- "The inclusion criteria were all adult Covid-19 patients with and/or without CKD". 

2) Kidney Disease Outcomes Quality Initiative (KDOQI) and  Kidney Disease Improving Global Outcomes (KDIGO) guidelines define CKD as the presence of kidney damage or decreased renal function for more than 3 months. Was this considered in the diagnosis of CKD in this study?

3) Section 3.1- "Post-CKD: patients diagnosed by the doctor..." . Is there more to this paragraph? If not, this sentence seems odd.

4) Table 2. What exactly does "Cholesterol" mean?

5) 3.4 title is the same as 3.3

6) Table 4-Please state that Exp (B) is the odds ratio.

7) Figure 2- what does kind of CKD mean?

8) Instead of stating the p-value as 0.000, report the actual p-value (this can be obtained in SPSS) or say it as <0.0001

Author Response

REVIEWER 1

Open Review

English language and style

( ) English very difficult to understand/incomprehensible
( ) Extensive editing of English language and style required
( ) Moderate English changes required
(x) English language and style are fine/minor spell check required
( ) I don't feel qualified to judge about the English language and style

Yes

Can be improved

Must be improved

Not applicable

Does the introduction provide sufficient background and include all relevant references?

(x)

( )

( )

( )

Are all the cited references relevant to the research?

(x)

( )

( )

( )

Is the research design appropriate?

( )

(x)

( )

( )

Are the methods adequately described?

( )

(x)

( )

( )

Are the results clearly presented?

( )

(x)

( )

( )

Are the conclusions supported by the results?

( )

(x)

( )

( )

Comments and Suggestions for Authors

This is a retrospective study that sought to identify factors that contribute to chronic kidney disease in Covid-19 patients admitted to two hospitals in Indonesia. While the study sample size and design seem to be satisfactory, the manuscript can be improved by addressing the points mentioned below:

1) The authors should clarify the inclusion criteria. This statement is rather confusing- "The inclusion criteria were all adult Covid-19 patients with and/or without CKD". 

Reply;

The reviewer has a point. Actually, we have wanted to investigate patients diagnosed with CKD by the doctor after or during SARS-CoV-infection in the ward. We have now removed the words “with and/or without CKD” to avoid confusion. Apology for this.

2) Kidney Disease Outcomes Quality Initiative (KDOQI) and Kidney Disease Improving Global Outcomes (KDIGO) guidelines define CKD as the presence of kidney damage or decreased renal function for more than 3 months. Was this considered in the diagnosis of CKD in this study?

Reply

Thank you for the valuable comment. Based on patients’ medical records, the doctors diagnosed the patients as having CKD in either stages 2, 3, 4 or 5. The diagnosis was based on the calculation of GFR based on the CKD Epidemiology Collaboration (CKD-EPI). This is the practice in our country.

3) Section 3.1- "Post-CKD: patients diagnosed by the doctor..." . Is there more to this paragraph? If not, this sentence seems odd.

Reply; Indeed, it was truncated. It should read “Contributing Factors to CKD Following Covid-19 Diagnosis”.

6) Table 4-Please state that Exp (B) is the odds ratio.

The information has been added. Thank you for being detailed with our paper.

7) Figure 2- what does kind of CKD mean?

Reply: Actually what we mean is the type of CKD. We have corrected this in the article accordingly. Thank you.

8) Instead of stating the p-value as 0.000, report the actual p-value (this can be obtained in SPSS) or say it as <0.0001

This has been done accordingly. Thank you very much for the valuable comment.

Reviewer 2 Report

I have reviewed with interest the manuscript entitled “Factors Contributing To Chronic Kidney Disease Following Covid-19 Diagnosis In Pre-Vaccinated Hospitalized Patients” authored by Ramatillah, D.L., et al to be considered for publication in  Vaccines.  The manuscript documents clinical biomarkers of bad prognosis and therapeutic observations among  adult patients admitted for COVID-19 with no previous history of chronic kidney disease. Authors documented the clinical deterioration of COVID-19  admitted patients that developed neuropathy and some other forms of co-morbidities. Although the manuscript describe some observations that might be of interest, it lacks the rigor of a clinical retrospective study. From the reading of the manuscript the concepts of association and causation are confounded. Furthermore, there are many parts of the manuscript where clinical I information is missing and misplaced. As such, I can not recommend the publication of the manuscript.

Major points

  1. From the abstract, there are mathematical ways to express numbers below 0.0005 and many observations are claimed to have statistical significance with a p=0.000. This is not acceptable. The manuscript would benefit from the revision of a clinical statistician to verify that the author’s claims can be quantitatively sustained. 
  2. The introduction has a lot of unsubstantiated speculations. Not every clinical observation can be substantiated with basic research. huCoV-2 entrance into cells and anti-hypertensive treatment are far from having a clear association. The speculation only confuses. By the way, the virus enters the body through the respiratory route and the host cell through a cell receptor. 
  3. From the reading of material and methods (2.1), it reads …”The exclusion criteria were patients with history of CKD prior to hospital admission, ….Then, how is it possible that a patient develop a chronic nephropathy (CKD according to authors) in less than 100 days (longer stay)? Wouldn’t be a more accurate diagnosis “acute kidney failure”? There are many markers of kidney function that are missing, albuminuria, glomerular filtration rate, imaging studies; as well as clinical indicators of kidney disease progression such as diuresis. It is simply poorly described.
  4. Table 1 does not describe demographic characteristics but vital signs. Are those at admittance?
  5. In table 2, factors are not described anywhere in the text. For example “Sepsis”, does it mean septic shock with bacteremia, systemic inflammatory response, bacteremia alone? I assume that due to the fact that many patients receive antibiotics some etiologic microorganism must have been identified. Another one ‘Cytokine storm” which cytokine(s)? What was the concentration to claim the patient was storming? All of the laboratory determination have normal ranges and the pathology is associated with values out of the range, what is the value? For instance, every single human being has cholesterol or uric acid in plasma. It is a value out of the normal range what determines the diagnosis and prognosis. This observation can be applied to table 4. 
  6. What were the used antibiotic doses? What is the mechanism of action of the used anti-virals?
  7. Figure 2. What is a “kind” of CKF? What does censored, non-censored means in this terms?

Minor comments

  1. No patient is admitted to the ICU before hospital admission.
  2. If alcohol intake is such a big problem among Indonesian males, then what is your casuistic in this retrospective study?
  3. No human being is an elder at 50 years of age. I checked Indonesian life expectancy, almost 72 years.

Author Response

Reviewer 2

Open Review

English language and style

( ) English very difficult to understand/incomprehensible
( ) Extensive editing of English language and style required
(x) Moderate English changes required
( ) English language and style are fine/minor spell check required
( ) I don't feel qualified to judge about the English language and style

Yes

Can be improved

Must be improved

Not applicable

Does the introduction provide sufficient background and include all relevant references?

( )

( )

(x)

( )

Are all the cited references relevant to the research?

(x)

( )

( )

( )

Is the research design appropriate?

( )

( )

(x)

( )

Are the methods adequately described?

( )

( )

(x)

( )

Are the results clearly presented?

( )

( )

(x)

( )

Are the conclusions supported by the results?

( )

( )

(x)

( )

Comments and Suggestions for Authors

I have reviewed with interest the manuscript entitled "Factors Contributing To Chronic Kidney Disease Following Covid-19 Diagnosis In Pre-Vaccinated Hospitalized Patients" authored by Ramatillah, D.L., et al to be considered for publication in  Vaccines.  The manuscript documents clinical biomarkers of bad prognosis and therapeutic observations among adult patients admitted for COVID-19 with no previous history of chronic kidney disease. Authors documented the clinical deterioration of COVID-19-admitted patients that developed neuropathy and some other forms of co-morbidities. Although the manuscript describes some observations that might be of interest, it lacks the rigor of a clinical retrospective study. From the reading of the manuscript, the concepts of association and causation are confounded. Furthermore, there are many parts of the manuscript where clinical I information is missing and misplaced. As such, I can not recommend the publication of the manuscript.

Major points

  1. From the abstract, there are mathematical ways to express numbers below 0.0005 and many observations are claimed to have statistical significance with a p=0.000. This is not acceptable. The manuscript would benefit from the revision of a clinical statistician to verify that the author's claims can be quantitatively sustained. 

Reply; We have revised p=0.000 to p < 0.0001 as pointed out by reviewer 1. Thank you very much for this comment.

  1. The introduction has a lot of unsubstantiated speculations. Not every clinical observation can be substantiated with basic research. huCoV-2 entrance into cells and anti-hypertensive treatment are far from having a clear association. The speculation only confuses. By the way, the virus enters the body through the respiratory route and the host cell through a cell receptor. 

Reply; Thank you for this point. Indeed, the virus enters the body through the respiratory route and the host cell through a cell receptor. We are aware that huCoV-2 entrance into cells and anti-hypertensive treatment are far from having a clear association still for now. Nevertheless, the information was quoted from frontiers as below:

Muhamad, S. A., Ugusman, A., Kumar, J., Skiba, D., Hamid, A. A., & Aminuddin, A. (2021). COVID-19 and Hypertension: The What, the Why, and the How. Frontiers in Physiology, 12(May), 1–11. https://doi.org/10.3389/fphys.2021.665064

Having said that, we have added the statement further on “The virus enters the body through the respiratory route and the host cell through a cell receptor. ” in the introduction to better clarify our points as suggested by the reviewer. Thank you for the valuable comment.

  1. From the reading of material and methods (2.1), it reads …" The exclusion criteria were patients with history of CKD prior to hospital admission, ….Then, how is it possible that a patient develop a chronic nephropathy (CKD according to authors) in less than 100 days (longer stay)? Wouldn't be a more accurate diagnosis "acute kidney failure"? There are many markers of kidney function that are missing, albuminuria, glomerular filtration rate, imaging studies; as well as clinical indicators of kidney disease progression such as diuresis. It is simply poorly described.

Reply; The reviewer may be right in saying that a more accurate diagnosis is AKI. Nevertheless, we need to explain our local scenario where currently in Indonesia, unfortunately, patients do not regularly go for a laboratory diagnosis and often they go undiagnosed. Therefore, it is rather common to see that CKF diagnosis is only done /detected during hospital admission.

Regarding some markers of kidney function that may be missing, we are limited by the fact that most hospitals in Indonesia do not conduct such intensive assessments as seen in other parts of the world including in first-world countries. Nevertheless, to our knowledge, our data on the markers of kidney function are one of the most comprehensive ones being reported locally. We did conduct GFR tests. Taking the reviewer’s point, we have included the fact that some markers such as imaging studies as well as clinical indicators of kidney disease progression such as diuresis have not been done in the study limitation.

  1. Table 1 does not describe demographic characteristics but vital signs. Are those at admittance?

      Reply: Thank you for pointing this out. Yes, they are vital signs during hospital admission. We have rephrased it to “Association between CKD and the socio-demographic and vital signs during admission” for better accuracy. Thank you for the valuable comment.

  1. In table 2, factors are not described anywhere in the text. For example "Sepsis", does it mean septic shock with bacteremia, systemic inflammatory response, or bacteremia alone?

Reviewer 2 is very detailed. Based on the medical record, the patients developed sepsis from bacteremia which progressed to SIRS (Systemic Inflammatory Response Syndrome). Therefore, the “sepsis” here tends to mean bacteremia as well. Since this is a retrospective study, the word “sepsis” was picked up from the doctor’s writing. We have included the information in the text accordingly, for better clarity. Thank you.

  1. I assume that due to the fact that many patients receive antibiotics some etiologic microorganism must have been identified. Another one is' Cytokine storm" which cytokine(s)? What was the concentration to claim the patient was storming? All of the laboratory determination have normal ranges and the pathology is associated with values out of the range, what is the value? For instance, every single human being has cholesterol or uric acid in plasma. It is a value out of the normal range what determines the diagnosis and prognosis. This observation can be applied to table 4. 

The doctors have made the diagnosis of cytokine storms based on the D-dimers values which we reported in our study in Table 3. Since this is a retrospective study based on patients’ medical records, we are depending on the doctors’ assessment 100%; also based on patients’ comorbidities and cannot report beyond this further.

Taken the reviewer’s comments, we have added the statement “Moreover, patient data retrieval is rather complicated in Indonesia since it is directly related to the quality of the data recording” to our study’s limitation as below. Thank you for the valuable comment.

Our study has some limitations in that the study duration is relatively short and the samples were from two sites (both from private hospitals in Jakarta). Additionally, laboratory parameters such as interleukin-6 (IL-6) and lactate dehydrogenase (LDH) that may serve as important prognostic markers in CKD were not measured due to lack of facilities, although the D-dimer laboratory test indicated that most patients have hypercoagulation. Moreover, patient data retrieval is rather complicated in Indonesia since it is directly related to the quality of the data recording.

  1. What were the used antibiotic doses? What is the mechanism of action of the used anti-virals?

Thank you for the valuable comment. We have now included all of the antibiotic doses in Table 6, accordingly.

The respective  mechanism of action of the anti-virals used has also been included below.

The active form of remdesivir acts as a nucleoside analog and inhibits the RNA-dependent RNA polymerase (RdRp) of coronaviruses including SARS-CoV-2. https://www.nature.com/articles/s41467-020-20542-0

Favipiravir is an anti-viral agent that selectively and potently inhibits the RNA-dependent RNA polymerase (RdRp) of RNA viruses. https://www.ncbi.nlm.nih.gov/pmc/articles/PMC5713175/

  1. Figure 2. What is a "kind" of CKF? What does censored, non-censored means in this terms?

Reply; Apology for the unconventional word. This has also been picked up by reviewer 1. We have corrected it to the type of CKD (Chronic Kidney Disease).

Subjects who have died, dropped out, or move out are not counted as “at risk” i.e., subjects who are lost are considered “censored” and are not counted in the denominator. The words are unique to Kaplan-Meier analysis and due to this fact, we have to maintain it. Thank you for the comment, nevertheless.

Reference; Goel MK, Khanna P, Kishore J. Understanding survival analysis: Kaplan-Meier estimate. Int J Ayurveda Res. 2010 Oct;1(4):274-8. doi: 10.4103/0974-7788.76794. PMID: 21455458; PMCID: PMC3059453.

Minor comments

  1. No patient is admitted to the ICU before hospital admission.

Reply. Apology. This is our mistake. What we meant here was that patients were already in a severe stage when they were admitted to the hospital. The sentence has been corrected accordingly and thanks for pointing this out.

  1. If alcohol intake is such a big problem among Indonesian males, then what is your casuistic in this retrospective study?

Reply; Here, we attempted to explain in the context of higher male predominance with CKD. In Indonesia, most males have bad lifestyles such as smoking and alcohol consumption that may contribute to lower immune response.

  1. No human being is an elder at 50 years of age. I checked Indonesian life expectancy, almost 72 years.

Reply. Done. This has been changed to “old”. Thanks for pointing this out.

Reviewer 3 Report

Introduction section

1- Please replace "acute renal failure" with "acute kidney injury" throughout the text.

2- Please add the following references about the association of COVID-19 with kidney involvement.

- Kordzadeh-Kermani E, Khalili H, Karimzadeh I. Pathogenesis, clinical manifestations and complications of coronavirus disease 2019 (COVID-19). Future Microbiol. 2020 Sep;15:1287-1305. doi: 10.2217/fmb-2020-0110. 

- Sabaghian T, Kharazmi AB, Ansari A, Omidi F, Kazemi SN, Hajikhani B, Vaziri-Harami R, Tajbakhsh A, Omidi S, Haddadi S, Shahidi Bonjar AH, Nasiri MJ, Mirsaeidi M. COVID-19 and Acute Kidney Injury: A Systematic Review. Front Med (Lausanne). 2022 Apr 4;9:705908. doi: 10.3389/fmed.2022.705908.

Method section

1- Please provide the definition used for CKD in the cohort. For example, "calculated GFR based on the CKD-EPI or MDRD formula" & "KDIGO criteria".  

2- Please determine whether the "RT-PCR COVID-19 test" of recruited patients was positive or not.

3- Please define Cholesterol (Hypercholestrolemia is a better term), Hyperuricaemia, Anemia, Cytokine storm, and Clinical outcome. 

4- Please determine whether laboratory values correspond to baseline (at the time of diagnosis) or during the course of the disease.

Result section

1- According to type of statistical analyses used, the term "correlation" should be replaced by "association" within this section.

2- Please determine the follow-up period of studied patients.

3- Please add data about the severity of stage of COVID-19 provided by the WHO (WHO criteria) in the cohort.

4-  Please list antiviral and antibacterial agents used in the cohort as a table.

5- If possible, please determine how many patients required dialysis or kidney transplantation. 

Discussion section

1- Statements about "Favipravir dose adjustment" seem unnecessary and they could be removed. 

2- Although Remdesivir is deemed to be a relatively nephrotoxic agent (compared to Favipravir), it should be taken into account that Remdisivir is usually given in patients with more severe COVID-19 that affected patients are more vulnerable to development of AKI  and consequently, CKD. In turn, Favipravir is mostly used in mild to moderate types of COVID-19. This issue should be discussed.

Author Response

REVIEWER 3

Open Review

English language and style

( ) English very difficult to understand/incomprehensible
( ) Extensive editing of English language and style required
(x) Moderate English changes required
( ) English language and style are fine/minor spell check required
( ) I don't feel qualified to judge about the English language and style

Yes

Can be improved

Must be improved

Not applicable

Does the introduction provide sufficient background and include all relevant references?

(x)

( )

( )

( )

Are all the cited references relevant to the research?

(x)

( )

( )

( )

Is the research design appropriate?

( )

(x)

( )

( )

Are the methods adequately described?

( )

(x)

( )

( )

Are the results clearly presented?

( )

(x)

( )

( )

Are the conclusions supported by the results?

(x)

( )

( )

( )

Comments and Suggestions for Authors

Introduction section

1- Please replace "acute renal failure" with "acute kidney injury" throughout the text.

Reply: Done.  "Acute renal failure" has been replaced with "acute kidney injury" throughout the text. Thank you.

2- Please add the following references about the association of COVID-19 with kidney involvement.

- Kordzadeh-Kermani E, Khalili H, Karimzadeh I. Pathogenesis, clinical manifestations and complications of coronavirus disease 2019 (COVID-19). Future Microbiol. 2020 Sep;15:1287-1305. doi: 10.2217/fmb-2020-0110. 

- Sabaghian T, Kharazmi AB, Ansari A, Omidi F, Kazemi SN, Hajikhani B, Vaziri-Harami R, Tajbakhsh A, Omidi S, Haddadi S, Shahidi Bonjar AH, Nasiri MJ, Mirsaeidi M. COVID-19 and Acute Kidney Injury: A Systematic Review. Front Med (Lausanne). 2022 Apr 4;9:705908. doi: 10.3389/fmed.2022.705908.

Reply: Done. The references have been added accordingly. Thank you.

Method section

  • Please provide the definition used for CKD in the cohort. For example, "calculated GFR based on the CKD-EPI or MDRD formula" & "KDIGO criteria".

Reply: Those patients who were diagnosed with CKD after Covid-19 status were included in the report study, and the GFR was calculated based on CKD-EPI.

  • Please determine whether the "RT-PCR COVID-19 test" of recruited patients was positive or not.

Reply; Patients with RT-PCR COVID-19 test were positive.

  • Please define Cholesterol (Hypercholestrolemia is a better term), Hyperuricaemia, Anemia, Cytokine storm, and Clinical outcome. 

Reply; This was similarly pointed by reviewer 1. Yes, we have changed “cholesterol” to Hypercholestrolemia, accordingly. Thank you for pointing this out.

We have also included the below definitions accordingly in the text, for better clarity.

2.4 Definition

Hypercholesterolemia = high level of cholesterol in the blood

Hyperuricemia = high level of uric acid in blood

Anemia = low of hemoglobin in red blood cells

Cytokine storm = inflammatory disease caused by high cytokine levels in the body

Clinical outcome = the final result of the treatment (death or recover)

Sepsis = a 'life-threatening organ dysfunction caused by a dysregulated host response to infection

  • Please determine whether laboratory values correspond to baseline (at the time of diagnosis) or during the course of the disease.

Reply; Apology, it was after the diagnosis We have corrected the text as below:

2.3. Data Collecting and Handling

Data were arranged according to the socio-demographic status, laboratory test following Covid-19 diagnosis, and current medication and transcribed to the CRF.

Result section

  • According to type of statistical analyses used, the term "correlation" should be replaced by "association" within this section.

Reply; Done. The reviewer is good. Thanks for pointing this out. We have replaced all “correlation” with “association” which is indeed more accurate here.

  • Please determine the follow-up period of studied patients.

Reply; This is a retrospective study based on patient's complete medical record conducted at two Private Hospitals in Jakarta. Patients with RT-PCR COVID-19 test were recruited via a convenience sampling between March 2020 and September 2021. Thank you.

3- Please add data about the severity of stage of COVID-19 provided by the WHO (WHO criteria) in the cohort.

Reply: This has been added accordingly. Thank you for this.

  1. According to the WHO there is 3 categories stage of covid-19: [1] critical covid-19: in this category patients already develop acute respiratory distress syndrome (ARDS), sepsis, septic shock or the condition that normally required provision of life-sustaining therapies, [2] severe covid-19: oxygen saturation < 90% on room air, signs of pneumonia and signs of severe respiratory distress and [3] non-severe covid-19: absence of signs of severe or critical disease (27). However, according to the health minister of the republic of Indonesia there are 4 categories of severity covid-19: [1] asymptomatic, [2] moderate symptoms [patient with pneumonia symptoms and SpO2 93–95 %], [3] severe symptoms [patient with pneumonia and SpO2<93%] and [4] critical illness [patient with Acute Respiratory Distress Syndrome (ARDS), sepsis and septic shock (28). In this study, the disease severity is based on the local guideline from the Ministry of Health of Indonesia.

  1. Health Organization W. Guideline Clinical management of COVID-19 patients: living guideline, 18 November 2021 [Internet]. 2021 [cited 2022 Nov 14]. Available from: https://apps.who.int/iris/bitstream/handle/10665/349321/WHO-2019-nCoV-clinical-2021.2-eng.pdf
  2. Michael, Ramatillah DL. TREATMENT PROFILE AND SURVIVAL ANALYSIS ACUTE RESPIRATORY DISTRESS SYNDROME (ARDS) COVID-19 PATIENTS. International Journal of Applied Pharmaceutics. 2022;14(Special Issue 2). Post-CKD: patients diagnosed by the doctor after or during SARS-CoV-2 infection in the ward, and all patients who had history of CKD before being admitted to the hospital were excluded.

4) Table 2. What exactly does "Cholesterol" mean?

Reply; Apology, it was indeed “hypercholesterolemia”. Thanks for pointing this out as also pointed out by the other two reviewers.

5) 3.4 title is the same as 3.3

Reply; This has been corrected accordingly, Thank you for the scrutiny.

4- Please list antiviral and antibacterial agents used in the cohort as a table.

Reply; Although a painful effort, we have done it in Table 6 below accordingly, to enrich the data further. Thank you.

Favipiravir was the most used antiviral in this study (19.6%) and Levofloxacin was the most common antibiotic used in this study (Table 6).

Table 6. Prevalence and Percentage Antiviral and Antibiotic Used

No

Antiviral

N (%)

Antibiotic

N (%)

1

isoprinosine (3 x 500 mg)

5 (1.3)

Levofloxacin (1 x 750 mg)

66 (17.5)

2

favipiravir (day 1 2 x1600 mg day 2-4 2 x 600mg)

74 (19.6)

Meropenem (3 x 1 gram)

3 (0.8)

3

remdesivir (1 x 200 mg)

31 (8.2)

Ceftriaxone (1 x 2 gram)

13 (3.4)

4

isoprinosin (3 x 500 mg)

+ remdesivir (1 x 200 mg)

25 (6.6)

Azitromycin  (1 x 500 mg)

43 (11.4)

5

isoprinosin (3 x 500 mg)

+ favipiravir (day 1 2 x1600 mg day 2-4 2 x 600mg)

29 (7.7)

Levofloxacin (1 x 750 mg)+ Meropenem (3 x 1 gram)

17 (4.5)

6

isoprinosin (3 x 500 mg)

+ remdesivir (1 x 200 mg)

 + favipiravir (day 1 2 x1600 mg day 2-4 2 x 600mg)

21 (5.6)

Levofloxacin (1 x 750 mg)+ Ceftriaxone (1 x 2 gram)

37 (9.8)

7

isoprinosin (3 x 500 mg)

+ favipiravir (day 1 2 x1600 mg day 2-4 2 x 600mg)

+ remdesivir (1 x 200 mg)

+ oseltamivir (2 x 75 mg)

3 (0.8)

Meropenem (3 x 1 gram)

 + Azitomycin  (1 x 500 mg)

2 (0.5)

8

favipiravir (day 1 2 x1600 mg day 2-4 2 x 600mg)+ remdesivir (1 x 200 mg)

33 (8.7)

Azitromycin  (1 x 500 mg)

+ Ceftriaxone (1 x 2 gram)

11 (2.9)

9

isoprinosin (3 x 500 mg)

+ remdesivir (1 x 200 mg)

+ favipiravir (day 1 2 x1600 mg day 2-4 2 x 600mg)

1 (0.3)

Fosfomycin  (2 x 1 gram) + Ceftazdim

1 (0.3)

10

isoprinosin  (3 x 500 mg) + oseltamivir (2 x 75 mg)

+ invermectin  (1 x 24 mg)

1 (0.3)

Cotrimoxazole  (1 x 960 mg)

 + Ceftriaxone (1 x 2 gram)

2 (0.5)

11

No antiviral agent

57 (15.1)

Azitromycin  (1 x 500 mg)

+ Meropenem (3 x 1 gram)

+ Ceftriaxone (1 x 2 gram)

2 (0.5)

12

Isoprinosin  (3 x 500 mg)+ Remdesivir (1 x 200 mg)

 + Methisoprinol (3 x 500 mg)

1 (0.3)

Levofloxacin (1 x 750 mg)+ Ceftriaxone  (1 x 2 gram)+ Meropenem (3 x 1 gram)

5 (1.3)

13

Oseltamivir (2 x 75 mg)

+ Favipiravir (day 1 2 x1600 mg day 2-4 2 x 600mg)

+ Remdesivir (1 x 200 mg)

18 (4.8)

Azitromycin  (1 x 500 mg)

+ Meropenem (3 x 1 gram)

 + Levofloxacin (1 x 750 mg)

14 (3.7)

14

Isoprinosin  (3 x 500 mg)+ Favipiravir (day 1 2 x1600 mg day 2-4 2 x 600mg)

+ Oseltamivir (2 x 75 mg)

2 (0.5)

Azitromycin  (1 x 500 mg)

+ Cefotaxime  (3 x 1 gram) + Meropenem (3 x 1 gram)

2 (0.5)

15

Ivermectin + Remdesivir (1 x 200 mg)

1 (0.3)

Levofloxacin (1 x 750 mg) + Ceftriaxone + Meropenem (3 x 1 gram)

 + Amikacin  (2 x 500 mg)

1 (0.3)

16

Isoprinosin  (3 x 500 mg)+ Remdesivir (1 x 200 mg)+ Oseltamivir (2 x 75 mg)

3 (0.8)

Fosfomycin  (2 x 1 gram)+ Doripenem (3 x 1 gram)+ Cefixime  (2 x 200 mg) + Vancomycin  (1 x 1,5 gram)

 + Levofloxacin (1 x 750 mg)

1 (0.3)

17

Isoprinosin  (3 x 500 mg)+ Remdesivir (1 x 200 mg)

 + Favipiravir (day 1 2 x1600 mg day 2-4 2 x 600mg) + Invermectin  (1 x 24 mg)

2 (0.5)

Azithromycin  (1 x 500 mg)

+ Levofloxacin (1 x 750 mg) + Ceftriaxone (1 x 2 gram)

13 (3.4)

18

Methisoprinol (3 x 500 mg) + Favipiravir (day 1 2 x1600 mg day 2-4 2 x 600mg)

1 (0.3)

Azithromycin  (1 x 500 mg)

+ Ceftriaxone (1 x 2 gram) + Levofloxacin (1 x 750 mg)+ Cefixime  (2 x 200 mg)

1 ((0.3)

19

Favipiravir (day 1 2 x1600 mg day 2-4 2 x 600mg)

+ Oseltamivir (2 x 75 mg)

13 (3.4)

Ceftriaxone (1 x 2 gram) + Cefixime  (2 x 200 mg)

1 (0.3)

20

Oseltamivir (2 x 75 mg)

7 (1.9)

Ceftazidime (3 x 1 gram)

 + Cotrimoxazole  (1 x 960 mg)

 + Amikacin  (2 x 500 mg)+ Levofloxacin (1 x 750 mg)+ Azithromycin  (1 x 500 mg)

 + Ceftriaxone (1 x 2 gram) + Meropenem (3 x 1 gram)

1 (0.3)

21

Favipiravir (day 1 2 x1600 mg day 2-4 2 x 600mg) + Oseltamivir (2 x 75 mg)

+ Isoprinosin  (3 x 500 mg)

8 (2.1)

Azithromycin  (1 x 500 mg)

 + Cefotaxime  (3 x 1 gram)

3 (0.8)

22

Favipiravir (day 1 2 x1600 mg day 2-4 2 x 600mg) + Oseltamivir (2 x 75 mg)

+ Isoprinosin  (3 x 500 mg) + Remdesivir (1 x 200 mg)

32 (8.5)

Azithromycin (1 x 500 mg)

 + Levofloxacin (1 x 750 mg)

27 (7.1)

23

Oseltamivir (2 x 75 mg)+ Isoprinosin (3 x 500 mg)

9 (2.4)

Cefotaxime  (3 x 1 gram)

2 (0.5)

24

Oseltamivir (2 x 75 mg)+ Remdesivir (1 x 200 mg)

1 (0.3)

Lefofloxacin (1 x 750 mg)+ Ceftriaxone + Azithromycin  (1 x 500 mg)

+ Meropenem (3 x 1 gram)

4 (1.1)

25

Total

378

Levofloxacin (1 x 750 mg) + Ceftazidime (3 x 1 gram) + Cefotaxime  (3 x 1 gram)

1 (0.3)

26

Clindamycin (3 x 300 mg)+ Levofloxacin  (1 x 750 mg)

1 (0.3)

27

Cefotaxime  (3 x 1 gram)+ Cefixime  (2 x 200 mg)

1 (0.3)

28

No Antibiotic

29 (7.7)

29

Azithromycin  (1 x 500 mg)

 + Cefoperazone (2 x 2 gram)

8 (2.1)

30

Azithromycin  (1 x 500 mg)

 + Cefoperazone (2 x 2 gram)+ Ceftazidim

2 (0.5)

31

Azithromycin  (1 x 500 mg)

 + Cefoperazone (2 x 2 gram)+ Ceftriaxone (1 x 2 gram) + Meropenem (3 x 1 gram)

2 (0.5)

32

Azithromycin  (1 x 500 mg)

 + Cefoperazone (2 x 2 gram)+ Meropenem (3 x 1 gram)

6 (1.5)

33

Azithromycin  (1 x 500 mg)

 + Cefoperazone (2 x 2 gram)+ Meropenem (3 x 1 gram)

+ Ceftazidim (3 x 1 gram)

2 (0.5)

34

Azithromycin  (1 x 500 mg)

 + Levofloxacin (1 x 750 mg)+ Cefoperazone (2 x 2 gram)

4 (1.1)

35

Azithromycin  (1 x 500 mg)

 + Levofloxacin (1 x 750 mg) + Cefoperazone(2 x 2 gram) + Cefotaxime  (3 x 1 gram)

1 (0.3)

36

Azithromycin  (1 x 500 mg)

 + Levofloxacine  (1 x 750 mg)+ Cefoperazone (2 x 2 gram)+ Ceftriaxone (1 x 2 gram)

6 (1.6)

37

Azithromycin  (1 x 500 mg)

 + Levofloxacin  (1 x 750 mg)+ Cefoperazone (2 x 2 gram) + Meropenem (3 x 1 gram)

32 (8.5)

38

Meropenem (3 x 1 gram)+ Ceftazidime (3 x 1 gram)

1 (0.3)

39

Azithromycine  (1 x 500 mg)

 + Levofloxacine  (1 x 750 mg)+ Ceftazidime (3 x 1 gram)

1 (0.3)

40

Azithromycine  (1 x 500 mg)

 + Levofloxacine  (1 x 750 mg) + Cefoperazone (2 x 2 gram)+ Meropenem (3 x 1 gram)

 + Ceftazidime (3 x 1 gram)

1 (0.3)

41

Azithromycine  (1 x 500 mg)

 + Levofloxacine  (1 x 750 mg)+ Cefoperazone (2 x 2 gram) + Meropenem (3 x 1 gram) + Ciprofloxacine (2 x 500 mg)

1 (0.3)

42

Levofloxacine  (1 x 750 mg)+ Cefixime  (2 x 200 mg)+ Cefotaxime  (3 x 1 gram)

1 (0.3)

43

Levofloxacine (1 x 750 mg) + Cefoperazone (2 x 2 gram) + Ceftriaxone (1 x 2 gram)

1 (0.3)

44

Levofloxacine  (1 x 750 mg)+ Cefotaxime  (3 x 1 gram)

1 (0.3)

45

Levofloxacine  (1 x 750 mg)+ Ceftazidime (3 x 1 gram)

1 (0.3)

46

Levofloxacine  (1 x 750 mg)+ Ceftriaxone (1 x 2 gram) + Ceftazidime (3 x 1 gram)

1 (0.3)

47

Levofloxacine  (1 x 750 mg)+ Meropenem (3 x 1 gram)

 + Ceftazidime (3 x 1 gram)

1 (0.3)

48

Levofloxacine  (1 x 750 mg)+ Meropenem (3 x 1 gram)

 + Ciprofloxacine (2 x 500 mg)

1 (0.3)

49

Total

378

5- If possible, please determine how many patients required dialysis or kidney transplantation. 

Reply: Thanks for this comment. No patients require dialysis in this study. There was a single patient who might require but he did not make it to the dialysis schedule.

Discussion section

1- Statements about "Favipravir dose adjustment" seem unnecessary and they could be removed. 

Reply: Done. This part has been removed accordingly.

2- Although Remdesivir is deemed to be a relatively nephrotoxic agent (compared to Favipravir), it should be taken into account that Remdisivir is usually given in patients with more severe COVID-19 that affected patients are more vulnerable to development of AKI  and consequently, CKD. In turn, Favipravir is mostly used in mild to moderate types of COVID-19. This issue should be discussed.

Reply: Thanks fot this. We have discussed it in more detail as below:

Favipiravir is an anti-viral agent that selectively and potently inhibits the RNA-dependent RNA polymerase (RdRp) of RNA viruses (59).

Besides favipiravir, remdesivir is also used in treating of covid-19 infection. Although remdesivir is deemed to be a relatively nephrotoxic agent and usually is administered to patients with more severe COVID-19 infection, affected patients are more vulnerable to the development of AKI. According to Wu et al, administration of remdesivir is not recommended in patients with eGFR below 30 ml/min (60). In fact, administration of remdesivir for more than five days may worsen AKI (60, 61). Overall, the active form of remdesivir acts as a nucleoside analog and inhibits the RNA-dependent RNA polymerase (RdRp) of coronaviruses including SARS-CoV-2 (62).

  1. Wu B, Luo M, Wu F, He Z, Li Y, Xu T. Acute Kidney Injury Associated With Remdesivir: A Comprehensive Pharmacovigilance Analysis of COVID-19 Reports in FAERS. Front Pharmacol. 2022 Mar 25;13.
  2. Goldman A, Bomze D, Dankner R, Hod H, Meirson T, Boursi B, et al. Cardiovascular adverse events associated with hydroxychloroquine and chloroquine: A comprehensive pharmacovigilance analysis of pre‐COVID‐19 reports. Br J Clin Pharmacol. 2021 Mar 22;87(3):1432–42.

Round 2

Reviewer 2 Report

The problem of the paper in my opinion are not clerical mistakes but a faulty clinical design. Therefore, in my opinion the manuscript is still not acceptable to recommend its publication.

Author Response

Thank you for your comments.

Reviewer 3 Report

- In the method section, please provide more subjective and scientific definitions with relevant references for "Hypercholesterolemia, Hyperuricemia, Anemia, Cytokine storm, and Sepsis".

- In the discussion section, please add and briefly discuss the results of the following articles about remdesivir nephrotoxicity:

1 - Gérard AO, Laurain A, Fresse A, Parassol N, Muzzone M, Rocher F, et al. Remdesivir and Acute Renal Failure: A Potential Safety Signal From Disproportionality Analysis of the WHO Safety Database. Clinical pharmacology and therapeutics. 2021;109(4):1021-4. https://doi.org/10.1002/cpt.2145.

2- Silva NAO, Zara A, Figueras A, Melo DO. Potential kidney damage associated with the use of remdesivir for COVID-19: analysis of a pharmacovigilance database. Cadernos de saude publica. 2021;37(10):e00077721. https://doi.org/10.1590/0102-311x00077721.

3- Wu B, Luo M, Wu F, He Z, Li Y, Xu T. Acute Kidney Injury Associated With Remdesivir: A Comprehensive Pharmacovigilance Analysis of COVID-19 Reports in FAERS. Frontiers in pharmacology. 2022;13:692828. https://doi.org/10.3389/fphar.2022.692828.

4- Izcovich A, Siemieniuk RA, Bartoszko JJ, Ge L, Zeraatkar D, Kum E, et al. Adverse effects of remdesivir, hydroxychloroquine and lopinavir/ritonavir when used for COVID-19: systematic review and meta-analysis of randomised trials. BMJ open. 2022;12(3):e048502. 

Author Response

Open Review

English language and style

( ) English very difficult to understand/incomprehensible
( ) Extensive editing of English language and style required
( ) Moderate English changes required
(x) English language and style are fine/minor spell check required
( ) I don't feel qualified to judge about the English language and style

Yes

Can be improved

Must be improved

Not applicable

Does the introduction provide sufficient background and include all relevant references?

(x)

( )

( )

( )

Are all the cited references relevant to the research?

(x)

( )

( )

( )

Is the research design appropriate?

(x)

( )

( )

( )

Are the methods adequately described?

( )

(x)

( )

( )

Are the results clearly presented?

(x)

( )

( )

( )

Are the conclusions supported by the results?

( )

(x)

( )

( )

Comments and Suggestions for Authors

Comment 1

- In the method section, please provide more subjective and scientific definitions with relevant references for "Hypercholesterolemia, Hyperuricemia, Anemia, Cytokine storm, and Sepsis".

Response:

Hypercholesterolemia = excessively high plasma cholesterol levels, and is a strong risk factor for many negative cardiovascular events (27).

Hyperuricemia = the presence of high level of uric acid in blood (28).

Anemia = hemoglobin (Hb) concentration and/or red blood cell (RBC) numbers which is lower than normal and is insufficient to meet the physiological needs of an individual (29, 30).

Cytokine storm = a general term applied to the maladaptive cytokine release in response to infection or other stimuli (31).

Sepsis = a medical emergency that describes the body’s systemic immunological response to an infectious process that may lead to the development of end-stage organ dysfunction and death (32).

Comment 2

- In the discussion section, please add and briefly discuss the results of the following articles about remdesivir nephrotoxicity:

1 - Gérard AO, Laurain A, Fresse A, Parassol N, Muzzone M, Rocher F, et al. Remdesivir and Acute Renal Failure: A Potential Safety Signal From Disproportionality Analysis of the WHO Safety Database. Clinical pharmacology and therapeutics. 2021;109(4):1021-4. https://doi.org/10.1002/cpt.2145.

2- Silva NAO, Zara A, Figueras A, Melo DO. Potential kidney damage associated with the use of remdesivir for COVID-19: analysis of a pharmacovigilance database. Cadernos de saude publica. 2021;37(10):e00077721. https://doi.org/10.1590/0102-311x00077721.

3- Wu B, Luo M, Wu F, He Z, Li Y, Xu T. Acute Kidney Injury Associated With Remdesivir: A Comprehensive Pharmacovigilance Analysis of COVID-19 Reports in FAERS. Frontiers in pharmacology. 2022;13:692828. https://doi.org/10.3389/fphar.2022.692828.

4- Izcovich A, Siemieniuk RA, Bartoszko JJ, Ge L, Zeraatkar D, Kum E, et al. Adverse effects of remdesivir, hydroxychloroquine and lopinavir/ritonavir when used for COVID-19: systematic review and meta-analysis of randomised trials. BMJ open. 2022;12(3):e048502. 

Response:

Administration of remdesivir is not recommended in patients with eGFR below 30 ml/min (66-68). In some studies (67, 68), using remdesivir for covid-19 treatment has been reported to increase the risk of developing AKI as compared to using other drugs such as tocilizumab, hydroxychloroquine and lopinavir/ritonavir. In fact, administration of remdesivir for more than five days have been reported to worsen AKI (66, 69) except for a single study by Izcovich et al, where remdesivir has not been reported to pose a significant risk for AKI when used as covid-19 treatment (70). Overall, the active form of remdesivir acts as a nucleoside

Round 3

Reviewer 3 Report

All new comments have been implemented and the current revised manuscript can be considered for publication. 

Author Response

Thank you for your comments